# Gold Nanoclusters-Based NIR-II Photosensitizers with Catalase-like Activity for Boosted Photodynamic Therapy

**DOI:** 10.3390/pharmaceutics14081645

**Published:** 2022-08-07

**Authors:** Qing Dan, Zhen Yuan, Si Zheng, Huanrong Ma, Wanxian Luo, Li Zhang, Ning Su, Dehong Hu, Zonghai Sheng, Yingjia Li

**Affiliations:** 1Department of Medicine Ultrasonics, Nanfang Hospital, Southern Medical University, Guangzhou 510515, China; 2Shenzhen Institute of Advanced Technology, Chinese Academy of Sciences, Shenzhen 518055, China; 3Faculty of Health Sciences, University of Macau, Taipa, Macau SAR, China

**Keywords:** photodynamic therapy, NIR-II fluorescence, gold nanoclusters, hypoxia, nanozyme, cancer theranostics

## Abstract

Photodynamic therapy (PDT) under fluorescence imaging as a selective and non-invasive treatment approach has been widely applied for the therapy of cancer and bacterial infections. However, its treatment efficiency is hampered by high background fluorescence in the first near-infrared window (NIR-I, 700–900 nm) and oxygen-dependent photosensitizing activity of traditional photosensitizers. In this work, we employ gold nanoclusters (BSA@Au) with the second near-infrared (NIR-II, 1000–1700 nm) fluorescence and catalase-like activity as alternative photosensitizers to realize highly efficient PDT. The bright NIR-II fluorescence of BSA@Au enables the visualization of PDT for tumor with a high signal-to-background ratio (SBR = 7.3) in 4T1 tumor-bearing mouse models. Furthermore, the catalase-like activity of BSA@Au endows its oxygen self-supplied capability, contributing to a five-fold increase in the survival period of tumor-bearing mice receiving boosted PDT treatment compared to that of the control group. Moreover, we further demonstrate that BSA@Au-based PDT strategy can be applied to treat bacterial infections. Our studies show the great potential of NIR-II BSA@Au as a novel photosensitizer for boosted PDT against cancer and bacterial infections.

## 1. Introduction

Photodynamic therapy (PDT) is acknowledged as a clinical treatment modality for cancer and bacterial infections, which is selective, non-invasive, and feasible to combine with imaging. It generally employs a visible or near-infrared laser to excite photosensitizers to generate fluorescence (FL) for guiding therapy. Meanwhile, it generates singlet oxygen (^1^O_2_) in the presence of oxygen to kill cancer cells and bacteria [1,2,3,4]. To date, organic dyes (i.e., porphyrins, indocyanine dyes, BODIPYs, and aggregation-induced emission dyes) [5,6,7,8], inorganic nanoparticles (i.e., quantum dots, black phosphorus, and TiO_2_) [9,10,11], polymer nanoparticles (i.e., polyfluorene and polythiophene) [12,13], and other nanomaterials (i.e., noble metal complexes and metallic nanomaterials) [14,15], have been developed for PDT. However, the FL emission of these photosensitizers is limited in visible (400–700 nm) and the first near-infrared (NIR-I, 700–900 nm) windows, resulting in intravital imaging with high background signal. Moreover, the ^1^O_2_ generation of photosensitizers gradually decreases within tissue due to the pre-existing hypoxia and consumption of oxygen during the PDT process, leading to unsatisfactory PDT outcomes [16]. Therefore, it is of great significance to develop novel photosensitizers with long emission wavelength and self-supplied oxygen for boosted PDT against tumors and infectious diseases.

In recent years, several dyes with photosensitive activity in the second near-infrared (NIR-II, 1000–1700 nm) window have been explored, which exhibits unique merits over the NIR-I window, such as a reduction in photon absorption and scattering, as well as improvement of signal-to-background ratio (SBR) [17,18,19]. However, these NIR-II photosensitizers generally show low PDT efficiency due to the oxygen insufficiency within laser irradiation area. Recently, NIR-II gold nanoclusters (NCs), consisting of several to about a hundred gold atoms with a size below 3 nm, are of particular interest to researchers owing to their intrinsic FL emission, renal clearance, and high biocompatibility [20]. Gold NCs with the NIR-II emission have been developed for the bright in vivo NIR-II FL imaging of malignant tumors, bones, and brain vessels [21,22,23,24,25]. In addition to FL imaging capability, gold NCs could also generate toxic ^1^O_2_ under NIR laser irradiation [26]. Furthermore, our previous studies reported that gold NCs showed catalase-like activity, which could decompose endogenous hydrogen peroxide (H_2_O_2_) within tumor tissue to O_2_ for enhancing PDT [27]. However, to our best knowledge, there has no reports on the utilization of the three features of gold NCs, including NIR-II FL, ^1^O_2_ generation, and catalase-like activity, for boosted PDT against tumors and bacterial infections. 

Herein, we developed the NIR-II gold NCs (BSA@Au) as novel photosensitizers for boosted PDT. As shown in Figure 1, BSA@Au was synthesized by one-step biomineralization. The facile synthesis process is advantageous for biological applications. The bright NIR-II FL of BSA@Au could realize high-performance in vivo bioimaging for the optimization of PDT. Meanwhile, the superb catalase-like activity of BSA@Au could catalyze the decomposition of excess H_2_O_2_ to produce O_2_ within the microenvironment of the tumor and bacterial-infected tissues for enhancing PDT. Our studies provide a paradigm of using gold NCs as efficient photosensitizers for boosted PDT.

## 2. Materials and Methods

### 2.1. Materials

Gold (III) chloride trihydrate (HAuCl_4_·3H_2_O), bovine serum albumin (BSA), sodium borohydride (NaBH_4_), and 1,3-Diphenylisobenzofuran (DPBF) were provided by Sigma-Aldrich. The Cell counting kit-8 (CCK-8) was supplied by Dojindo. Pimonidazole HCl and IgG_1_ mouse antibody (conjugated with FITC) were purchased from Hypoxyprobe.

### 2.2. Synthesis of BSA@Au 

BSA@Au was prepared adapted from previous reports [28]. Briefly, all glassware were soaked with aqua regia (HCl:HNO_3_, volume ratio 1:3), then washed with ultrapure water. HAuCl_4_ solution (10 mM, 1.25 mL) was mixed with BSA solution (0.76 mM, 2.5 mL). Then, NaOH solution (1 M, 0.25 mL) and NaBH_4_ (0.1 M, 0.1 mL) were introduced. The whole process was under vigorously stirring. When the mixture turned into deep brown, it was incubated in a shaker (180 rpm, 37 °C) for 12 h to obtain BSA@Au nanoclusters. The as-prepared BSA@Au was purified by dialysis (molecular weight cutoff: 8–14 KD) with 0.1 M phosphate solution (pH ≈ 7.4) and stored in the dark at 4 °C for further investigation. 

### 2.3. In Vivo FL Imaging 

Mice bearing a 4T1 tumor received an intravenous (i.v.) injection of BSA@Au (0.46 mM, 200 µL) and were imaged at 0, 2, 4, 10, and 24 h. At 10 h, the main organs (including heart, liver, spleen, lung, and kidneys) and tumor were collected for ex vivo FL imaging. NIR-I FL imaging was conducted with excitation wavelength at 675 nm, and emission wavelength at 800 nm. The NIR-II FL imaging conditions were set as follows: excitation wavelength: 808 nm; long-pass filter: 1000 nm; and exposure time: 800 ms. 

### 2.4. In Vivo PDT for Tumor 

To evaluate the PDT efficacy, we randomly divided Balb/C mice bearing 4T1 tumor into 4 groups (n = 5): (1) Control; (2) Laser; (3) BSA@Au; (4) BSA@Au + Laser. At 10 h post i.v. injection of BSA@Au, the 808 nm laser (0.3 W/cm^2^, 30 min) was applied and the temperature of tumors in group (4) was recorded to avoid the thermal effect. Since the single dose treatment, the tumor volumes and body weights of mice were recorded every 2 days for further analysis. When tumor grew to ~800 mm^3^, the mouse was defined as dead to obtain the survival curve.

### 2.5. In Vitro Antibacterial Assay

For antibacterial experiments, the amounts of live MRSA after PDT treatment were quantified by plate counting method. Firstly, MRSA suspensions were mixed with PBS, BSA@Au, and H_2_O_2_ (200 μM). Then, the suspensions (OD_600nm_ ≈ 0.1) were placed in 96-well plates. After incubation at 37 °C for 30 min, the samples were further illuminated by 808 nm laser (0.3 W/cm^2^, 30 min). Then, the suspensions were diluted (1:1 × 10^5^) and transferred to LB agar plates for further cultivation at 37 °C. After 24 h, the colonies were counted by ImageJ software. In addition, bacterial morphology details of MRSA after different treatments were observed by scanning electron microscopy (SEM). 

### 2.6. In Vivo PDT for MRSA Infection 

To investigate in vivo antimicrobial performance, the infected mice were randomly divided into 4 groups (n = 5): (1) Control; (2) Laser; (3) BSA@Au; (4) BSA@Au + Laser. Mice of groups (3) and (4) were locally treated with BSA@Au solution (2.5 mM, 20 μL) at wound sites. The wound areas and body weights were recorded every 3 days. After 12 days of treatments, the wounds were excised for H&E staining analysis. 

### 2.7. Statistical Analysis

Data are shown as the mean ± SD. Statistical significance was analyzed by student’s *t*-test analysis. *p* < 0.05 was considered as statistical significance.

## 3. Results and Discussion

### 3.1. Preparation and Characterization of NIR-II BSA@Au

In this work, NIR-II emissive BSA@Au was synthesized by the biomineralization method. BSA was used as a template and protector for the growth of BSA@Au, and NaBH_4_ was introduced as the critical reductant to modulate the NIR-II FL emission intensity. As shown in the high-resolution transmission electron microscope (HRTEM) image, the core size of BSA@Au was ~2 nm (Figure 1A), which was below the glomerular filtration cutoff (~6 nm) for renal clearance after i.v. injection [29,30]. The hydrodynamic diameter of BSA@Au was 10.1 ± 1.4 nm and stayed highly stable after four weeks of storage in water or fetal bovine serum (FBS) (Figure 1B,C). The size of BSA@Au measured by dynamic light scattering (DLS) was larger than that determined by HRTEM because DLS detected an average hydrodynamic size whereas the HRTEM images exhibited the dehydration morphology of BSA@Au [31].

Generally, gold-based nanoparticles exhibit FL emission in both NIR-I and NIR-II windows. We used the NIR FL spectrum instrument and NIR-II FL imaging system to verify the FL property of BSA@Au. When prepared under various conditions, it demonstrated different fluorescent performance. We noted that the NIR-II FL intensity of BSA@Au was closely related to the molar ratio of BSA: HAuCl_4_ and NaOH: NaBH_4_, which were optimized to be 0.15 and 25, respectively. Particularly, BSA@Au did not show NIR-II FL emission without NaBH_4_ (Appendix A, Figure 1D,E and Appendix A), which indicated the important reduction role of NaBH_4_ to transfer Au^3+^ to Au to form the gold nucleus [32,33]. Furthermore, it was observed that NIR-II BSA@Au were formed rapidly within 0.5 h after adding NaBH_4_, and the FL intensity gradually became stronger with the reaction time, reaching the peak value at 12 h, before decreasing (Appendix A). Therefore, BSA (0.76 mM, 2.5 mL), HAuCl_4_ (10 mM, 1.25 mL), NaOH (1 M, 0.25 mL), and NaBH_4_ (0.1 M, 0.1 mL) were applied to synthesize NIR-II emissive BSA@Au under 37 °C for 12 h for further study.

To obtain deep insights into the optical properties of BSA@Au, its spectra properties were investigated. As shown in Figure 1F, the UV–Vis absorption spectrum of BSA@Au showed strong absorption at ~280 nm and decayed exponentially towards ~1100 nm. Moreover, the fluorescence emission spectrum exhibited two characteristic emission peaks at ~960 nm and ~1010 nm, reaching the NIR-II region. The mechanism of NIR-II emission of BSA@Au might involve complex electron transfer between BSA and gold cores [34]. In addition, we obtained the FL spectra of BSA@Au under various excitation wavelengths (450~808 nm). It showed an excitation wavelength-dependent FL emission peak, and the optimized FL emission spectrum was under 808 nm excitation (Appendix A). Such a broadband-excitable bright FL emission was beneficial for diverse applications, and 808 nm excitation was chosen for further investigation. 

Then, we used IR-26 as the reference and measured the fluorescent QY of the BSA@Au. Results showed that the QY of BSA@Au was measured to be ~3.5% (Appendix A), which was much higher than the QY of other NIR-II gold NCs (Table 1) and enabled it to realize bright FL imaging performance. Next, we tested the penetration depth of BSA@Au. Firstly, in the NIR-I window, the Eppendorf tube containing BSA@Au could not be detected when covered by chicken tissues (>4 mm thickness). In contrast, bright images of BSA@Au were obtained in NIR-II window at a penetration depth of more than 10 mm with high SBR (Appendix A). This indicated the potential that BSA@Au could achieve high-performance in vivo imaging. Next, we assessed the photostability of BSA@Au compared with the clinical NIR dye indocyanine green (ICG). It was found that, under NIR-II FL imaging, BSA@Au was still bright under 808 nm laser irradiation (0.3 W/cm^2^, 30 min) whereas the FL intensity of its ICG counterpart nearly decreased to ~0%, demonstrating that BSA@Au exhibited excellent photostability in contrast with ICG (Appendix A), which was owing to the role of BSA to protect BSA@Au from bleaching [35]. Taken together, the as-synthesized NIR-II emissive BSA@Au with a size of ~2 nm showed good water solubility, hydrodynamic size stability, photostability, and high QY of ~3.5%, demonstrating promising biomedical applications. 

### 3.2. Catalase-like Activity and Singlet Oxygen Generation

Tumor and infection microenvironments generally show high H_2_O_2_ concentration and low O_2_ level [39,40]. We hypothesized that the as-prepared BSA@Au could serve as a catalase mimic to alleviate hypoxic microenvironment within tumor and infected tissues (Figure 2A). In a typical experiment, after BSA@Au was added into H_2_O_2_ solution, we observed obvious gas bubbles generated from the mixed solution (Figure 2B). Furthermore, when BSA@Au reacted with H_2_O_2_ monitored by an ultrasound imaging system, it showed significantly enhanced imaging performance both on B mode and contrast-enhanced ultrasound mode (Figure 2C). More interestingly, clear gas echo signals were observed in 4T1 tumor-bearing mice injected with BSA@Au under an ultrasound imaging system (Appendix A), demonstrating that BSA@Au could catalyze H_2_O_2_ to produce O_2_.

To gain deep understanding of the catalase-like activity of BSA@Au, we investigated the apparent kinetic parameters according to Michaelis-Menten method and Lineweaver-Burk equation [41]. We found that O_2_ production was dependent on the concentration of BSA@Au and H_2_O_2_ (Figure 2D,E). Hence, it was believed that with more BSA@Au accumulation at the tumor sites or bacteria-infected lesions, more O_2_ would generate in situ. In Appendix A, the critical enzyme kinetic parameters of the Michaelis-Menten constant (K_m_) and maximum initial velocity (V_max_) were obtained and compared with other nanozymes. In this study, the apparent K_m_ reflected that BSA@Au showed high affinity towards H_2_O_2_, and V_max_ indicated the good catalytic ability of BSA@Au, which were superior to other nanozymes (Appendix A). However, compared with our previous study, the V_max_ of BSA@Au was lower than that of Au NCs-ICG (with a size of ~1 nm) by an order of magnitude, which might be attributed to the relatively larger size and less turnover number of BSA@Au [27]. Moreover, in the future, the size, shape, and ligand effects should be considered to improve the catalytic performance of BSA@Au.

Next, we evaluated the ^1^O_2_ generation ability of BSA@Au using 1,3-Diphenylisobenzofuran (DPBF) probe under 808 nm irradiation. Compared with the clinical photosensitizer ICG (^1^O_2_ generation efficiency, Φ_ΔICG_ = 0.2%), the Φ_ΔBSA@Au_ was determined to be about 0.4% (Figure 2F), which was 2.2-fold higher than that of ICG. More inspiringly, our results showed that the generated O_2_ from H_2_O_2_ degradation could enhance the generation of ^1^O_2_, which could overcome the pre-existing hypoxia and O_2_ depletion during the PDT process. In brief, upon 808 nm laser irradiation, the ^1^O_2_ generation of BSA@Au was measured by DPBF. As shown in Figure 2G, the DPBF was quenched by BSA@Au combining with 808 nm laser irradiation, and significantly decreased after the addition of H_2_O_2_. This suggested that BSA@Au could serve as a self-sufficient O_2_ supplying photosensitizer within H_2_O_2_-rich regions and thus enhance in vivo PDT for cancer and bacteria-infected diseases.

### 3.3. Toxicity Evaluation and In Vitro PDT

The cytotoxicity of BSA@Au was investigated using the CCK-8 assay. The 4T1 cells were incubated with BSA@Au of various concentrations. Compared with the control groups, there was almost no obvious cell viability decrease in 4T1 cells treated with BSA@Au (Figure 3A), which indicated the low cytotoxicity of BSA@Au. Similarly, BSA@Au exhibited no harm to red blood cells in hemolysis assay (Figure 3B). Based on the above results, we confirmed the negligible in vivo toxicity of BSA@Au. Two weeks after i.v. injection of the BSA@Au with high concentration (5 mM, 100 µL), the blood samples and main organs, including heart, liver, spleen, lung, and kidneys, were collected from mice for blood routine examinations and H&E analysis, respectively. This showed that the tissue sections and blood indicators of BSA@Au group were not significantly altered compared to normal references (Appendix A). The superior biocompatibilities of the BSA@Au might be attributed to the natural carrier of BSA and the ultrasmall size of BSA@Au, which together supported the in vivo application of BSA@Au for imaging and therapy.

Given the high ^1^O_2_ generation ability of BSA@Au, we tested the PDT efficacy against 4T1 cells. As shown in Figure 3C, Calcein-AM and PI dyes were used to visually evaluate the live/dead 4T1 cells after PDT. Nearly no red FL signals were detected in the control, BSA@Au, and Laser groups, while intense red FL signals in the BSA@Au + Laser group were observed. Cell viability after PDT was further quantified by CCK-8 method, revealing that only ~43.7% cells survived when treated by BSA@Au and laser irradiation. However, control groups did not show obvious damage to 4T1 cells (Figure 3D). These results demonstrated the efficient PDT potential of BSA@Au.

### 3.4. In Vivo Imaging

Inspired by in vitro FL imaging performance of BSA@Au, the in vivo NIR-I and NIR-II FL imaging of BSA@Au was evaluated in 4T1 tumor-bearing mice. As illustrated in Figure 4A, images in the NIR-I window roughly outlined the tissue, due to the absorption and scattering of NIR-I photons by biological tissues. However, the imaging quality was well improved in the NIR-II window, showing the superiority of NIR-II FL imaging. Clearly, the FL signals of the tumor region were gradually increased with time and reached the accumulation peak at 10 h post-injection, and then gradually declined. The SBR was quantitatively analyzed. The tumor FL signal shows about 7.3-fold higher than that of background at 10 h post injection (Figure 4B), which could precisely discriminate and localize the tumor tissues. In addition, the metabolism and biodistribution of BSA@Au in major organs and tumor were investigated by ex vivo NIR-II FL imaging. As shown in Figure 4C and Appendix A, at 10 h post i.v. injection, the heart, lung, spleen, and kidneys exhibited nearly no or weak FL signals, while the liver and tumor showed strong FL intensity, suggesting the BSA@Au preferred to accumulate in these organs. The effective accumulation of BSA@Au in tumor could reduce the nanoparticles injection dosage to minimize the system hazards. Furthermore, different metabolic behaviors of BSA@Au in liver and kidneys were observed. The liver gradually exhibited FL signals within 10 h post injection of BSA@Au, while the kidneys showed bright FL signals at 0.5 h and 4 h post-injection, and the signal became weak at 10 h (Appendix A). In addition, the signal of tumor and main organs (heart, liver, spleen, lung, and kidneys) remarkably decreased at 48 h post injection (Appendix A). This suggested that BSA@Au could be cleared out from liver and kidneys because of the small size. Taken together, these important features enabled BSA@Au to be a promising medical probe for in vivo imaging to guide therapy.

### 3.5. In Vivo Anti-Tumor Therapy 

Due to the excellent ^1^O_2_ generation ability and catalytic activity, we explored the anti-cancer potential of BSA@Au on 4T1 tumor-bearing mice. Firstly, we evaluated the hypoxia modulation performance of BSA@Au. After i.v. injection of BSA@Au, the obtained immunofluorescence images of tumor sections showed that the hypoxic areas of tumor tissue markedly decreased from ~46.2% to ~28.2% (Figure 5A,B). This revealed that BSA@Au had desirable catalase-like activity to decompose the overexpressed H_2_O_2_ within the tumor region to produce O_2_ for alleviating the hypoxic tumor microenvironment.

Subsequently, we investigated the in vivo PDT outcomes of BSA@Au. Mice bearing 4T1 tumor were divided into four groups with random (n = 5): (1) Control; (2) Laser; (3) BSA@Au; (4) BSA@Au + Laser. For group (4), 10 h after i.v. injection of BSA@Au, the mice were irradiated by 808 nm laser (0.3 W/cm^2^, 30 min). Notably, all mice from groups (2)–(4) only received a single-dose injection and/or single-dose laser irradiation during treatment. Moreover, we monitored the temperature changes of irradiated regions during the treatment. No obvious thermal increase was observed (Appendix A), which suggested that only the PDT effect (without photothermal effect) was involved in the killing of cancer cells. The tumor volume variation and body weights were recorded every 2 days. Results showed that the tumor growth was significantly inhibited in the group (4) whereas the tumor volumes increased quickly in groups (1)–(3), indicating the superior antitumor efficiency of BSA@Au in combination with laser irradiation (Figure 6C and Appendix A). Based on the efficient anti-cancer ability, mice from BSA@Au + Laser group survived for a much longer period (60 days) compared with the other three groups (Figure 5D).

Meanwhile, tumor tissues from all groups were collected on the next day after various treatments for H&E staining and TUNEL assay. Obvious cell apoptosis and necrosis were found in BSA@Au + Laser group; in contrast, nearly no abnormality was detected in Control, Laser, and BSA@Au groups (Figure 5E). These results further showed that single-dose BSA@Au injection combined with single laser irradiation possessed great damage to cancer tissues. In addition, during the whole therapy process, no body fluctuation was recorded (Appendix A), indicating the high biosafety of BSA@Au combined with laser for PDT. The results together demonstrated that BSA@Au could remarkably modulate hypoxia and realize efficient in vivo anti-tumor outcomes under NIR-II FL imaging guidance.

### 3.6. In Vitro Antibacterial Efficacy 

Bacterial infection have been considered as a key element of cancer management [42]. Given that the infection microenvironment is similar to tumor microenvionment, which features low O_2_ and high expression of H_2_O_2_ [39,40,43], we investigated the anti-bacterial capability of BSA@Au by boosted PDT effect. MRSA, a known type of drug-resistant bacterium, was involved for further evaluation. Inspired by previous reports that human serum albumin has a unique affinity with MRSA [44], we investigated the binding capability of the BSA@Au towards MRSA. Tube 1 was set as the control group, only containing BSA@Au solution; in tubes 2 and 3, MRSA was incubated with PBS or BSA@Au for 4 h, respectively. The three samples were imaged under the NIR-II FL imaging system after centrifugation (5000 rpm, 5 min). The intense signal in tube 3 displayed that BSA@Au could tightly bind with MRSA. Notably, the FL signal of the suspensions was from free BSA@Au solutions, and the FL signal of the precipitates was from the BSA@Au binding with MRSA because BSA@Au could not be spun down at a low centrifugation speed unless they were bound with MRSA (Figure 6A). Then, we evaluated the in vitro anti-bacterial efficacy of BSA@Au in combination with 808 nm laser irradiation. MRSA suspensions were divided into eight groups: (1) Control; (2) Laser; (3) BSA@Au; (4) H_2_O_2_; (5) H_2_O_2_ + Laser; (6) BSA@Au + H_2_O_2_; (7) BSA@Au + Laser; and (8) BSA@Au + H_2_O_2_ + Laser. Compared with other treatments, the group (7) killing~70% of the bacteria because of the good toxic ^1^O_2_ generation ability of BSA@Au (Figure 6B, C). Considering the hypoxic microenvironment within infectious tissues, which lacks O_2_ and overexpresses H_2_O_2_, we added H_2_O_2_ (200 µM) to simulate the bacteria-infected microenvironment. Excitingly, no live bacteria colonies (~0%) was observed in group (8), revealing enhanced antibacterial effect after O_2_ enrichment within infectious microenvironment. More visibly, as shown in the SEM images, the MRSA cells remained intact with grape-like morphology in groups (1)–(6). However, the MRSA bacteria were distorted and denatured in group (7)–(8). Moreover, the obvious shrinking and cracking of bacteria in group (8) were recorded (Figure 6D) due to ^1^O_2_ generated from BSA@Au and H_2_O_2_ under 808 nm laser irradiation.

### 3.7. In Vivo Antibacterial Efficacy

To further verify the in vivo anti-bacterial effect, we established a MRSA-infected wound model on Balb/C mice, and then recorded the wound closure and bacterial residue after different treatments. At first, we observed the in vivo NIR-II FL imaging of MRSA-infected wound by local injection of BSA@Au. The bright FL signal was collected, and the SBR was determined to be about 66.8 (Appendix A), demonstrating the potential of BSA@Au as an NIR-II FL probe for in vivo imaging. The wounds treated by BSA@Au + Laser showed a significantly faster healing speed than those of the group (1)–(3), and almost completely healed within 12 days, whereas the wound areas of control group were still about 85.5% of the original wounds (Figure 7A,B). In addition, at the end of the treatment period, we collected the wound tissues from healthy mouse, control group, and BSA@Au + Laser group for continuing cultivation in LB culture medium. As expected, a large number of live MRSA cells were observed in the control group, whereas no bacteria remained in the healthy mouse and BSA@Au + Laser groups because of the excellent anti-bacterial effect of the treatment (Figure 7C). The skin tissues were also used for histology analysis. H&E staining assay displayed that obvious effusion, loose dermis, and inflammatory cells were found in the infected tissue, whereas less necrosis, fewer granulated cells, and more fibrous tissues were observed in the cured wound (Figure 7D). During the whole treatment, no obvious changes in body weight were observed, suggesting PDT for MRSA therapy was safe and biocompatible (Appendix A). Moreover, during the PDT process, the skin temperature did not increase (Appendix A), which excluded the photothermal effect. MRSA-infection as a drug-resistant infectious disease showed little response to clinical antibiotics. However, PDT, as a novel treatment strategy for bacterial infections, could achieve the superb anti-MRSA effect and avoid treatment resistance without using antibiotics. Hence, in the future, BSA@Au might be utilized to detect and treat the deep-seated infections in NIR-II phototheranostic platform.

## 4. Conclusions

In summary, we provided proof-of-principle evidence for the application of an O_2_ self-sufficient photosensitizer for boosted cancer/infection PDT under the guidance of NIR-II FL imaging. In this system, BSA@Au was endowed with strong NIR-II FL emission, superb fluorescent QY (~3.5%), strong photobleaching resistance, excellent catalase-like activity, as well as good biocompatibility to realize imaging-guided PDT for cancer and bacteria-infected lesions. With bright NIR-II imaging performance, BSA@Au visualized the tumor location with high SBR (~7.3) and biodistribution, as well as the metabolism pathway itself, enabling high-performance imaging guided PDT. Moreover, BSA@Au served as a catalase mimic decomposing H_2_O_2_ to O_2_ for alleviating a hypoxic microenvironment to augment O_2_-dependent PDT. With single 808 nm laser irradiation, it achieved efficient tumor growth inhibition and MRSA-infection curation. Therefore, the developed BSA@Au might fulfill the unmet need for phototheranostics of cancer and bacterial infections.

## Data Availability

Data supporting reported results beyond what is reported in this manuscript are available upon reasonable request from the corresponding authors.

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
