# Peer review of "Gold Nanoclusters-Based NIR-II Photosensitizers with Catalase-like Activity for Boosted Photodynamic Therapy"

_pharmaceutics, 2022, doi:10.3390/pharmaceutics14081645_

Round 1

Reviewer 1 Report

The authors made a commendable effort to develop gold nanoclusters based theragnostic particles. I just have a few minor comments:

1.       Line 56, Please revise the sentence for more clarity

2.       The abbreviation should be written out in full in first use. Please check for all abbreviations.

3.       Please include the reduction in photon absorption and scattering as one of the major benefits of the NIR-II region.

4.       How was the concentration of BSA@Au measured?  Was the concentration of BSA considered the actual concentration of BSA@Au? How was the optimal concentration of BSA@AU determined for IV injection in mice?  The injection concentration and the various concentrations tested for cytotoxicity/hemolytic rate% are in different units. Can you please use the same unit for a better perspective?

5.       The major concern with this study is the accumulation of BSA@Au in the liver. Although the data demonstrate the clearance after 24 hours post IV injection, the accumulation of BSA@ Au for so long may elicit hepatotoxicity. The cytotoxicity assay should be performed with hepatic cells at higher concentrations at least for 24-48 hours.

Author Response

Dear Reviewer,

We have revised our manuscript as you suggested. Please see the attachment.

Best,

Yingjia Li

Reviewer 2 Report

The authors describe synthesis, characterization and evaluation of BSA coated gold nanoparticles in this study for possible application as imaging and treatment platform. While the study is interesting and important there are several concerns that have to be addressed before the manuscript can be considered for publication, these are listed below:

1.      In general, the manuscript addresses imaging and treatment for breast cancer and microbial infections which are totally different issues and better if presented separately. In its current form, it appears that the anti-microbial section of the manuscript is weak with relatively less discussion in the abstract and conclusion. Hence, it will be better to remove it and publish separately.

2.      The authors use phrases such as “..for the therapy of cancer and bacterial infections, which is generally visualized by fluorescence imaging” and “visualization of PDT” which are not clear. From what appears is that the authors are trying to correlate the relevance of imaging and PDT with the same formulation, however the way it is presented it is confusing.

3.      For the nanoparticle purification what was the MWCO for the dialysis tubing.

4.      What were the concentrations of H2O2 used in the experiments and how do they compare with in vivo levels observed in tumors?

5.      How does the treatment conditions (0.3 W/cm2 for 30 min) compare to the clinically approved light doses (fluence)? The light doses used appear to be significantly higher. Also, the imaging requires a significantly higher exposure time (800 ms), which might be because of the low absorption and quantum yield of these nanoparticles. The authors should discuss limitations of the system.

6.      The authors should provide a quantification for tissue distribution data provided in figure 4D.

7.      There are several instances where the language should be corrected. For example, “…and freshly ice-cold NaBH4 (0.1 M, 0.1 mL) were sequentially introduced”, “Interestingly, BSA@Au possessed molecular-like opti-220 cal features, exhibiting two…”, etc.

Author Response

(The authors gave the same response as above.)

Reviewer 3 Report

Student’s t-test analysis performed on data with normally distributions.  If data was not normal distributed – better used nonparametric Mann-Whitney test (aka Wilcoxon test) instead.

‘Results’ section should be separated with ‘Discussion’ section.

Better used tables/figures to list key-finding of results – instead of lengthy text descriptions.

Article is too lengthy – 18 pages (main text) – suggest to consolidate and high-light the key-points - shorten the article.

Word of ‘significant’ or ‘significantly’ need to support by performing statistical comparison and list with p-values in text. Without showing p-values – should just use ‘better’ instead.

Author Response

(The authors gave the same response as above.)
